# Molecular Biology and Pathological Process of an Infectious Bronchitis Virus with Enteric Tropism in Commercial Broilers

**DOI:** 10.3390/v13081477

**Published:** 2021-07-28

**Authors:** Ana P. da Silva, Ruediger Hauck, Sabrina R. C. Nociti, Colin Kern, H. L. Shivaprasad, Huaijun Zhou, Rodrigo A. Gallardo

**Affiliations:** 1Department of Population Health and Reproduction, School of Veterinary Medicine, University of California, Davis 1089 Veterinary Medicine Dr, 4008 VM3B, Davis, CA 95616, USA; apdasilva@ucdavis.edu; 2Department of Pathobiology and Department of Poultry Science, Auburn University, 302J Poultry Science Building, 260 Lem Morrison Dr, Auburn, AL 36849, USA; mrh0079@auburn.edu; 3Zootechnical Hygiene Laboratory, School of Animal Science and Food Engineering, University of Sao Paulo, 225 Duque de Caxias St, Pirassununga 13635-900, SP, Brazil; snociti@usp.br; 4Department of Animal Science, College of Agriculture, University of California, Davis, One Shields Ave, Davis, CA 95616, USA; colin.kern@gmail.com (C.K.); hzhou@ucdavis.edu (H.Z.); 5California Animal Health and Food Safety Laboratory System, Tulare Branch, 18760 Rd 112, Tulare, CA 93274, USA; hlshivaprasad@ucdavis.edu

**Keywords:** IBV, infectious bronchitis, variants, whole-genome sequencing, enteric tropism, runting-stunting syndrome

## Abstract

Infectious bronchitis virus (IBV) induces respiratory and urogenital disease in chickens. Although IBV replicates in the gastrointestinal tract, enteric lesions are uncommon. We have reported a case of runting-stunting syndrome in commercial broilers from which an IBV variant was isolated from the intestines. The isolate, CalEnt, demonstrated an enteric tissue tropism in chicken embryos and SPF chickens experimentally. Here, we determined the full genome of CalEnt and compared it to other IBV strains, in addition to comparing the pathobiology of CalEnt and M41 in commercial broilers. Despite the high whole-genome identity to other IBV strains, CalEnt is rather unique in its nucleotide composition. The S gene phylogenetic analyses showed great similarity between CalEnt and Cal 99. Clinically, vent staining was slightly more frequent in CalEnt-infected birds than those challenged with M41. Furthermore, IBV IHC detection was more evident and the viral shedding in feces was overall higher with the CalEnt challenge compared with M41. Despite underlying intestinal lesions caused by coccidiosis and salmonellosis vaccination, microscopic lesions in CalEnt-infected chickens were more severe than in M41-infected chickens or controls, supporting the enteric tropism of CalEnt. Further studies in SPF chickens are needed to determine the pathogenesis of the virus, its molecular mechanisms for the enteric tropism, and its influence in intestinal health.

## 1. Introduction

Infectious bronchitis virus (IBV) belongs to the Gammacoronavirus genus and mainly causes respiratory and urogenital disease in chickens. Although IBV replicates and persists in the cecal tonsils and has been isolated from duodenum and jejunum, intestinal lesions are rare [1,2,3]. Genetic variation of the viral spike (S) gene may lead to an increased susceptibility to proteolytic activation and to an improved ability to bind cell receptors, enhancing its infectivity. In addition, genomic changes modulate the virus entry and influence tissue tropism, persistence, virulence, and host range [4]. This variability explains why coronaviruses genetically and antigenically similar to IBV can cause different clinical outcomes. For example, the turkey coronavirus (TCoV) induces severe enteric disease by the viral replication in enterocytes of the jejunum and ileum [5,6,7]. Molecular differences between TCoV and IBV are mainly found in the S gene, in which homologies are as low as 33%, validating the divergent tissue tropism of these closely related viruses [8].

An enterotropic IBV strain has been described for causing intestinal lesions, mainly in the rectum, consisting of the desquamation of epithelial cells on the villi tips and congestion of the intestinal submucosa [9,10]. This virus was originally isolated from the intestines of chickens presenting with respiratory signs [11]. In another study, an IBV strain isolated from broilers with enteric and respiratory diseases was compared to an isolate from broilers that prompted only respiratory disease. Both isolates induced similar pathology after experimental infection; both viruses were detected in the intestines but their infection did not cause enteric lesions [12]. IBV was also one of several pathogens isolated from the intestines of commercial broilers presenting with runting-stunting syndrome. Infection with these IBV isolates alone or combined with other microorganisms induced a reduced body weight in experimentally infected chickens [13].

Previously, we have described the detection and isolation of an IBV variant from the intestines of brown broiler chickens showing runting-stunting signs. Interestingly, the virus was not retrieved from the kidneys or respiratory tract. The enteric isolate was named CalEnt and was isolated from the intestines of chicken embryos, but not from the chorioallantoic membrane. Additionally, CalEnt induced runting-stunting syndrome-like lesions after experimental oculonasal infection of specific-pathogen-free (SPF) chickens. The S1 gene sequence of CalEnt showed nucleic acid sequence identities of 93.8% to IBV California 99 (Cal 99) and of 85.7% to IBV Arkansas DPI (Ark DPI) [14]. The aim of the present investigation was to determine the full genome of CalEnt and compare its genomic identity to other IBV strains, as well as to compare the pathobiology of CalEnt to that of the respiratory strain M41 after a controlled infection of commercial broilers.

## 2. Materials and Methods

### 2.1. Viruses

IBV CalEnt and M41 strains were used in experimental challenges. Both IBV strains were grown and titrated in embryonated chicken SPF eggs (Charles River, CT, USA) using standard procedures [15,16]. The viral dose of IBV CalEnt was 2 × 10^4^ EID_50_ and the dose of IBV M41 was 2 × 10^5^ EID_50_ in a 200-µL inoculum. The challenge was performed either oculonasally or via crop gavage.

### 2.2. Whole Genome Sequencing

Intestines from embryos infected with CalEnt were homogenized with PBS and RNA was extracted from a volume of 100 μL using TRIzol (ThermoFisher, Waltham, MA, USA). DNA depletion was performed using the Turbo DNA-Free Kit (ThermoFisher, Waltham, MA, USA) followed by rRNA depletion using the Terminator 5′-Phosphate-Dependent Exonuclease (Epicentre Biotechnologies, Madison, WI, USA) as per the manufacturer’s instructions. The DNase reaction was ceased by adding EDTA at a concentration of 5 mmol/L. The RNA was purified using the QIAamp Viral RNA Mini Kit (QIAgen, Valencia, CA, USA) without the addition of the carrier RNA, and eluted to a final volume of 30 μL. The RNA quality was evaluated using a 2100 Bioanalyzer (Agilent, Santa Clara, CA, USA). cDNA libraries were prepared using the NEBNext Ultra Directional RNA Library Prep Kit for Illumina (New England Biolabs, Ipswich, MA, USA). Libraries were sequenced using the Illumina at 100 bp paired end. Quality control of the reads, adaptor trimming, merging of paired end reads, and assembly were performed using Geneious Prime 2021.1.1. The sequences were assembled using IBV Cal99 as a reference (accession number AY414485). A second round of reference-based assembly was performed using the consensus sequence obtained in the first round as a reference. For accuracy verification, de novo assembly was performed using the SPAdes assembler. The two assemblies shared 100% nucleotide identity. The consensus was uploaded to GenBank (accession number MW556742).

The obtained consensus sequence was aligned with several IBV variants and TCoV using the MAFFT plugin [17] in Geneious Prime 2020.1.1. Nucleotide sequences were used to calculate homologies and for phylogenetic analyses using the maximum likelihood method based on the GTRGAMMAI model with 1000 bootstraps in Geneious Prime with the RaxML plugin [18]. Homology matrices were calculated, and phylogeny was performed in whole genomes, full S gene sequences, and 751 bp fragments of the S gene bearing the S1 hypervariable region (nucleotide positions 20,368 to 21,119 of GenBank accession number MW556742).

The S gene of CalEnt was submitted to the basic local alignment search tool (BLAST, blast.ncbi.nlm.nih.gov, accessed on 20 July 2021) and the whole S gene from sequences showing the highest homologies was aligned using the MAFFT plugin [17] in Geneious Prime 2020.1.1. The alignment was uploaded in SimPlot, which compares the percent identity of the S gene of CalEnt to other sequences and plots the identities versus the nucleotide positions [19].

### 2.3. Experimental Design

Two hundred one-day-old commercial broilers were obtained from a commercial hatchery. The birds were vaccinated at the hatchery against coccidia and *Salmonella*, but not for IBV. The chickens were divided into 5 groups of 40 birds each (Table 1) and housed in BSL2 rooms at the Teaching and Research Animal Care Services at the University of California, Davis. Feed and water were provided ad libitum. All animal experimental procedures were approved by the University of California, Davis Institutional Animal Care and Use Committee (Approval No. 19092). In the first week of life, 2 birds from group 1, one bird from group 2, and one bird from group 3 deceased from unspecific reasons unrelated to IBV (Table 1).

At 10 days of age, blood was collected from 10 birds of each group via wing vein puncture and tested for maternal antibodies. When 14 days old, four groups were inoculated with 200 μL of either one of the viruses directly into crop using a buttoned cannula or via oculonasal route. The fifth group was left unchallenged as a negative control (Table 1).

Respiratory signs were evaluated at 4, 6, 10, and 14 dpi. Individual respiratory sign scores were recorded as 0 (no signs), 1 (mild nasal rales or upper respiratory tract sounds), 2 (moderate tracheal rales), or 3 (severe respiratory sounds audible from a 20 cm distance) by a blinded investigator [20]. On the same days, the presence or absence of vent stains due to diarrhea was recorded.

At one, 2, 4, 6, 10, and 14 dpi, tears and cloacal swabs were collected from five chickens per group for detection of IBV by RT-qPCR. Tears were collected using granules of pure sodium chloride to stimulate lachrymation. Five birds per group were euthanized by carbon dioxide inhalation at 4 and 14 dpi, and tracheas and small intestine samples collected during necropsies were submitted for histopathology and immunohistochemistry (IHC).

### 2.4. Serology

Sera collected at 10 days of age were tested for antibodies against IBV by ELISA (IDEXX Laboratories, Westbrook, ME, USA). S/P ratios were transformed into titers as per the manufacturer’s recommendations. The cutoff titer for positivity is 396.

### 2.5. RT-qPCR

RNA was extracted from tears and cloacal swabs in PBS using the QIAamp Viral RNA Mini Kit (QIAgen, Valencia, CA, USA) as described by the manufacturer. A probe-based RT-qPCR using primers qIBf (5′-ATA CTC CTA ACT AAT GGT CAA CAA TG-3′) and qIBr (5′-GGC AAG TGG TCT GGT TCA C-3′) and probe qIBs (5′-CAA GCC ACT GAC CCT CAC AAT AA-3′) was performed, targeting a 134 bp fragment of the IBV M gene in 44 cycles of denaturation and combined annealing/extension steps [21,22]. Viral load was expressed as the total number of total cycles (44)-Cq. Although not entirely precise, the 44-Cq measurement provides an estimate of the viral load in the specimen. The relative difference between viral load in tears and cloacal swabs was calculated as the difference of the mean viral loads.

### 2.6. Histopathology and Immunohistochemistry

Tissues were fixed in 10% neutral buffered formalin, embedded in paraffin, sectioned at 4 μm, mounted on glass slides, and stained with hematoxylin and eosin (H&E) stains. IBV in situ antigen detection was performed using IHC staining as previously described [23]. A combination of two monoclonal antibodies targeting the IBV matrix and spike proteins was used [24,25].

### 2.7. Statistical Analyses

Viral loads (represented as 44-Cq) were compared using ANOVA followed by Tukey’s multiple comparison tests. Respiratory sign scores, vent staining, and uniformities were compared using Kruskal–Wallis (for non-parametric data) followed by Dunn’s multiple comparison tests. Data were analyzed using Prism 9 (GraphPad, La Jolla, CA, USA).

## 3. Results

### 3.1. Whole Genome Analyses

The complete genome homologies revealed that the IBV strains most closely related to CalEnt were Cal99 (94.3%), Cal 56b (94.2%), M41 (94.1%), and ArkDPI (94.1%) (Appendix A). The whole genome comparisons demonstrate that the highest genetic variability is in the S gene, where the nucleotide homologies to CalEnt were the lowest and averaged 81.1% (Table 2). Considering the full S gene, CalEnt was most similar to Cal 99 (88.4%), followed by Ark DPI (84.6%), Cal 1995 (83.4%), and Cal 56b (83.2%) (Appendix A). When focusing on the hypervariable region of the S1 gene, CalEnt shows a strikingly high homology to Cal 99 (95.5%) (Appendix A).

Despite the similarities to Cal 99, Cal 56b, M41, and Ark DPI, CalEnt did not cluster with other IBV strains in the whole genome phylogenetic analysis (Figure 1A). The full S gene phylogenetic tree shows CalEnt clustering with other strains that belong to genotype I, lineage 9 [26], but still relatively distant from its counterparts (Figure 1B), indicating its unique nucleotide composition. The S1 gene hypervariable region phylogeny corroborates the high homology between CalEnt and Cal 99, demonstrating that these two strains cluster closely together (Figure 1C).

The recombination analysis suggests that the S gene from CalEnt is the result of recombination events between Cal99, GA/1476/15, and IBS037A/14 (Figure 2). The initial portion of the CalEnt S gene (1–1675 bp) bears the S1 gene and shows high homology to Cal99. This region is commonly used for IBV genotyping and has hypervariable regions [26]. At nucleotide position 1327, a shift between Cal99 and GA/1476/15 occurs. From nucleotide position 2702 to 3094, IBS037A/14 is the most similar sequence to CalEnt found in GenBank using BLAST. Similarly, a small fragment of approximately 300 bp (nucleotide position 3094–3329) shows similarity to GA/1476/15, the highest homology on this fragment with BLAST.

### 3.2. Maternal Antibodies

IBV maternal antibodies were detected in 10% of the collected serum samples (*n* = 50) at 10 days of age. The geometric mean titer and standard deviation was 172 ± 166.

### 3.3. Clinical Signs

Overall, M41-infected birds presented with more severe respiratory clinical signs than birds infected with CalEnt at all time points (4, 6, 10, and 14 dpi) (*p* < 0.05, Figure 3). Within the M41-infected groups, at 4 dpi, birds inoculated oculonasally presented higher respiratory sign scores than birds inoculated via crop gavage. In contrast, birds infected with M41 via crop presented with the highest respiratory sign score at 14 dpi (*p* < 0.05, Figure 3).

Vent staining was observed more frequently in CalEnt-infected birds throughout the experiment, although no statistical differences were observed between IBV-infected birds (Figure 4). Birds challenged with CalEnt oculonasally appeared to present diarrhea more frequently than other groups at 6 dpi (*p* < 0.05, Figure 4). The unchallenged control birds also presented stained vents at 6, 10, and 14 dpi; however, the frequency was significantly lower than in IBV-challenged groups (*p* < 0.05, Figure 4).

### 3.4. Viral Shedding

In general, the viral load in tears was higher in M41-challenged birds than in those challenged with CalEnt. At 1, 2, and 4 dpi, the highest viral load was seen in birds challenged with M41 oculonasally; at 6 dpi and onward, the viral load in tears was higher in birds challenged with M41 via crop (*p* < 0.05). With the exception of swabs collected at 1 dpi, the viral loads in cloacal swabs were overall higher in CalEnt-infected chickens, although only statistically significant at 2 and 10 dpi (*p* < 0.05). Figure 5 shows the difference between the average viral load in tears and cloacal swabs. Starting at 2 dpi, the viral load difference is above zero in M41-infected birds, indicating higher viral shedding in the upper respiratory tract. In contrast, the viral load difference in chickens challenged with CalEnt is below zero, denoting higher viral elimination in feces in these birds. Statistical differences between viral loads are represented in Appendix A.

### 3.5. Histopathology and Immunohistochemistry

At 4 dpi, CalEnt-challenged birds presented with a mild to moderate lymphoplasmacytic infiltration of the tracheal mucosa, while the lesions in M41-infected birds were scored as moderate to severe. The IHC showed a small amount of virus antigen present in the tracheas of CalEnt-infected chickens, while the amount of IBV antigen in M41-infected birds was moderate. The presence of tracheal lesions was irrespective of the inoculation route. IBV viral antigen was not detected by IHC in the tracheas of any birds at 14 dpi.

Intestinal lesions were observed in all groups, including the unchallenged controls (group 5). A few birds in each group had small numbers of coccidia. At 4 dpi, mild to moderate IBV IHC staining was present in the cytoplasm of enterocytes at the tips of the villi of the duodenum and ileum from birds challenged with CalEnt (Figure 6A). In contrast, none to small amounts of viral antigen was observed in the intestines of M41-challenged birds at 4 dpi. At 14 dpi, mild to moderate virus staining was detected in the cytoplasm of lymphocytes present in the intestinal lamina propria of birds challenged with CalEnt (Figure 6B). No IHC staining was observed in M41-infected chickens or unchallenged controls.

## 4. Discussion

Although IBV is typically known for inducing respiratory and urogenital disease, few IBV strains have been described as being mainly enterotropic [9,10,11,12]. The IBV CalEnt strain has been associated with a case of runting-stunting in 14-day-old broiler chicks [14]. Despite all the efforts in characterizing the pathobiology of IBV strains with supposed enteric tropism, information on the molecular aspects of such strains are scarce. Here, we provide a comprehensive genomic characterization of CalEnt and a glimpse into the pathological process of this particular IBV strain in commercial broiler chickens.

The complete genomic sequence of CalEnt showed homologies higher than 94% to Cal 99, Cal 56b, M41, and Ark DPI (Appendix A). Despite this relatively high whole-genome identity to known and established IBV strains, CalEnt is still rather unique in nucleotide composition, and branches separately from other IBV strains phylogenetically (Figure 1A). When the phylogenetic and identity analyses were restricted to the most variable part of the viral genome—the S gene—the similarity between CalEnt and Cal 99 becomes more evident (Figure 1B,C). When narrowing down the analysis to the hypervariable region of the S gene, a strikingly high identity between CalEnt and Cal 99 was observed (Appendix A). It is noteworthy that, in this instance, the S1 hypervariable region was less variable than the remainder of the S gene. To further investigate possible evolutionary origins of the S gene of CalEnt, we performed recombination analyses using sequences that presented high homologies to CalEnt in BLAST. The S1 hypervariable region of CalEnt was most similar to Cal99, corroborating our phylogenetic findings (Figure 1C and Figure 2). The remainder of the S gene is mostly similar to GA/1476/15, an isolate retrieved from 11-day-old broilers from Georgia. It is noteworthy that GA/1476/15 was isolated from fecal samples in 2015 [27], while CalEnt was isolated from intestines in 2012. A small portion of the CalEnt S gene is similar to IBS037A/14, a Malaysian variant that is highly prevalent in broilers and layers and has been associated with nephropathogenic disease in outbreaks in 2014 and 2015 [28]. Although IBS037A/14 has never been detected in North America, an IBV recombinant sequence with similarity to this isolate has been described in Canada [29]. The S gene is the main determinant of tissue tropism and host specificity [30], and is therefore the most relevant gene for the genotyping of IBV strains. Altogether, these findings demonstrate evolutionary patterns of IBV and highlight the importance of the S gene in genotyping and pathotyping, most likely affecting tissue tropism.

Considering other genes, CalEnt also showed very high identity to other IBV isolates commonly used in vaccines in California, such as Mass, Conn, and Ark strains (Table 2). These results suggest that CalEnt is likely an IBV variant that arose from continued wild-type virus mutations in the presence of vaccine strains. The emergence of IBV variants, such as CalEnt in 2012 [14] and Cal 99 in 1999 [31,32], likely resulted from vaccine mishandling in the field, either by poor application techniques or by usage of vaccines at a lower dosage than recommended by the manufacturer. Vaccination issues may lead to rolling reactions and back-passaging of vaccine viruses within the flock, providing opportunities for mutations and recombination events that generate novel variants over time that might circumvent vaccine immunity [33].

Clinically, the commercial broilers challenged with either M41 or CalEnt in our experiment presented with respiratory signs and lesions, regardless of the route of inoculation or IBV strain, although respiratory disease was more severe in M41-infected birds than those challenged with CalEnt (Figure 3). A similar effect was observed in other studies investigating IBV strains isolated from the intestines, in which birds initially present respiratory signs [9,12]. In addition, the viral load in tears was overall higher in M41-infected birds (Figure 5, Appendix A), suggesting the viral shedding of M41 in the upper respiratory tract is higher than that of CalEnt. In contrast, vent staining was slightly more frequent in CalEnt-infected birds (Figure 4). Furthermore, the intestinal microscopic lesions and IBV IHC detection were more severe and the viral shedding in feces was overall higher with the CalEnt challenge compared with M41 (Figure 5, Appendix A). However, the M41 challenge dose was 10 times higher than that of CalEnt, due to difficulties in amplification of the wild-type CalEnt strain in SPF eggs. Nevertheless, the high frequency of diarrhea and the high shedding in feces are still noteworthy; perhaps these observations would have been more noticeable if the viral challenge was higher. An interesting observation is that, at 14 dpi, all IBV-infected groups were still shedding the virus via feces, which supports the existing knowledge that IBV persists in cecal tonsils for longer periods, making this tissue a good specimen for the detection and isolation of IBV [2].

Commercial broilers vaccinated against coccidia and *Salmonella* at the hatchery were used in this study, and some baseline intestinal damage might have been caused by these live vaccines. Intestinal health in commercial poultry is extremely complex and involves numerous factors, including attenuated microorganisms provided in vaccines, the microbiota, co-infections with other pathogens, and the nutritional contents of the feed [34]. In addition, a previous study using CalEnt failed to demonstrate the binding of the viral S1 protein to the tracheal or intestinal epithelia, suggesting that CalEnt might have a lower affinity to the host cell than other IBV strains in vitro [35]. It is possible that other factors present in the enteric milieu of commercial chickens (i.e., microbiome and tissue-specific proteases) facilitate the CalEnt binding to epithelial cells. By using commercial broiler chickens, our goal was to mimic a realistic poultry farm setting to understand the performance of the two IBV strains in the presence of other microorganisms. Despite the confounding lesions, possibly caused by the coccidia and *Salmonella* vaccines, the unchallenged control birds presented with less severe intestinal lesions compared to IBV-infected birds, and no IBV IHC staining was observed. The microscopic injuries observed in the CalEnt-infected birds appeared slightly more severe than those present in M41-infected birds, and the IBV IHC staining was more evident, supporting the enteric tropism of CalEnt as previously reported [14].

Regarding the inoculation route, the crop gavage inoculation was performed in order to investigate if fecal–oral infection would have a different impact on CalEnt pathogenesis as opposed to the known airborne transmission of IBV. In general, it appeared that the crop gavage inoculation led to a late onset of clinical disease that persisted for a prolonged period compared with the oculonasal inoculation. It is possible that a portion of the inoculum was inactivated in the proventriculus or ventriculus of challenged chickens, reducing the viral load and consequently delaying the clinical signs and viral shedding.

## 5. Conclusions

This study corroborates the evidence that CalEnt has a tropism for the intestinal tract, which is unusual for IBV. Its comprehensive pathogenesis and pathobiology in SPF chickens, its influence in intestinal health, and the underlying molecular mechanisms of this preferred replication site remain to be determined.

## Figures and Tables

**Figure 1 viruses-13-01477-f001:**
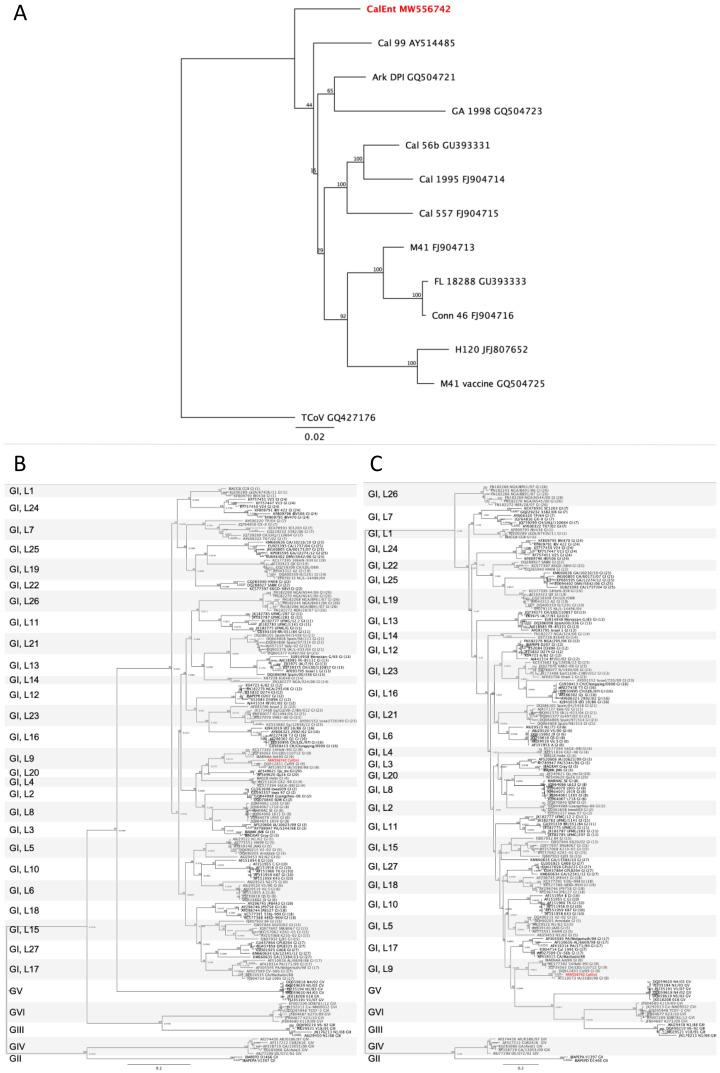
Phylogenetic analyses of the whole genome (**A**), the full S gene (**B**) and the hypervariable region of the S gene (**C**) of infectious bronchitis virus isolates. The maximum likelihood method with 1000 bootstrap replicates was used. The IBV CalEnt sequence is in red. G = genotype, L = lineage.

**Figure 2 viruses-13-01477-f002:**
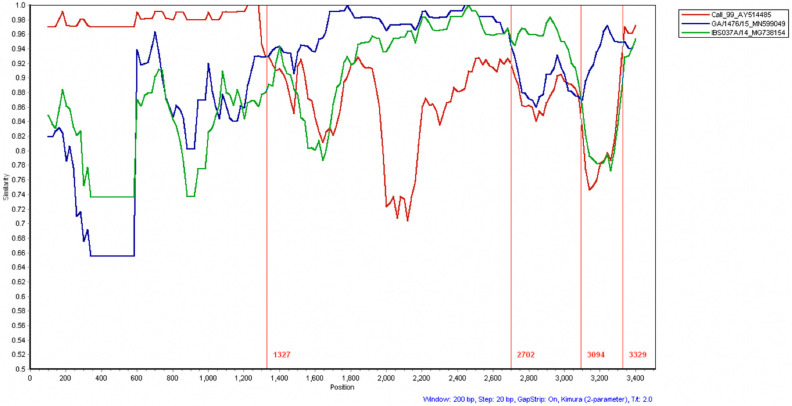
Similarity plot comparing the S gene of CalEnt (reference) to the S gene of Cal99, GA/1476/15, and IBS037A/14. The *y* axis represents the percent similarity to CalEnt, and the *x* axis represents the nucleotide position of the S gene. The vertical red lines represent the nucleotide position in which there is a sequence crossover.

**Figure 3 viruses-13-01477-f003:**
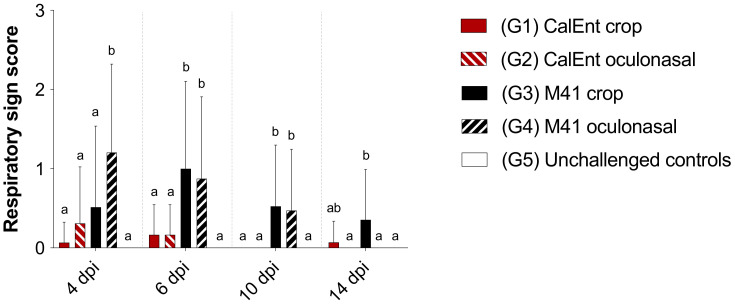
Respiratory sign scores of commercial broilers challenged with either IBV M41 or CalEnt oculonasally or via crop gavage. Different superscripts represent statistical significance (*p* < 0.05). Same letters represent no statistical difference (*p* > 0.05). Groups were compared using Kruskal–Wallis (for non-parametric data) followed by Dunn’s multiple comparison tests.

**Figure 4 viruses-13-01477-f004:**
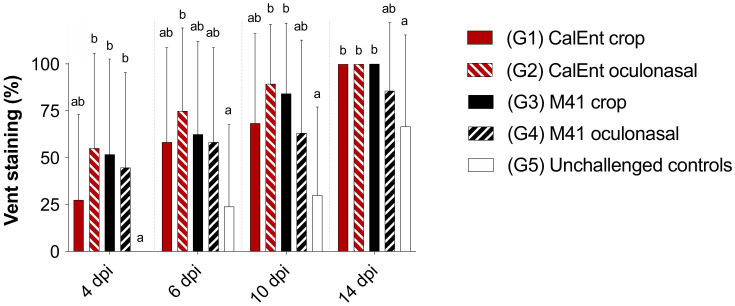
Percentage of commercial broilers presenting vent staining after a challenge with either IBV M41 or CalEnt oculonasally or via crop gavage. Different superscripts represent statistical significance (*p* < 0.05). Same letters represent no statistical difference (*p* > 0.05). Groups were compared using Kruskal–Wallis (for non-parametric data) followed by Dunn’s multiple comparison tests.

**Figure 5 viruses-13-01477-f005:**
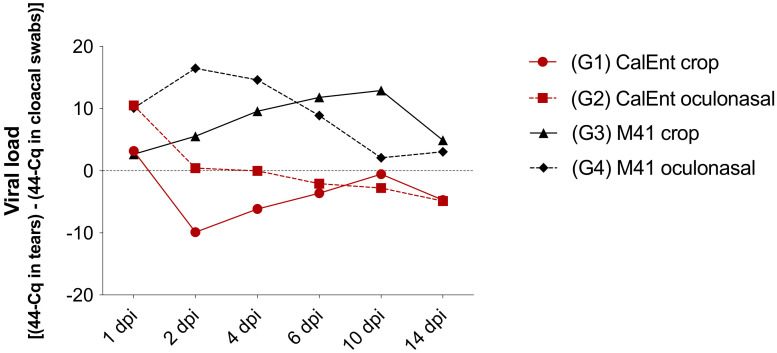
Difference between viral load in tears and viral load in cloacal swabs collected from commercial broilers challenged with either IBV M41 or CalEnt oculonasally or via crop gavage. Numbers above zero represent a higher viral load in tears. Numbers below zero represent a higher viral load in cloacal swabs. Group 5 (unchallenged controls) is not represented because no IBV amplification was observed in any sample at any timepoint.

**Figure 6 viruses-13-01477-f006:**
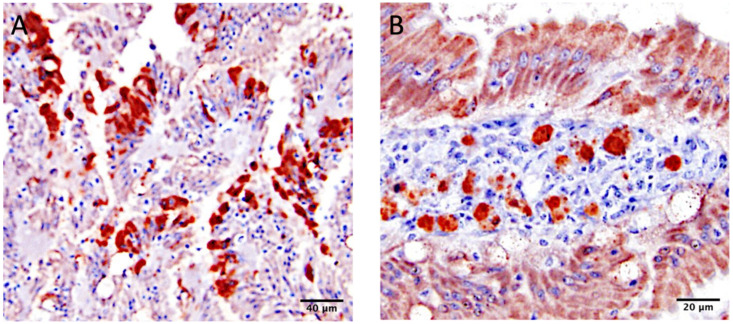
Infectious bronchitis virus antigen detection in intestinal sections of broilers challenged with CalEnt by immunohistochemistry (IHC): (**A**) IBV IHC staining in the cytoplasm of enterocytes from CalEnt-infected chickens at 4 dpi via oculonasal route. Scale bar = 40 µm; (**B**) IBV IHC staining in the cytoplasm of lymphocytes of the intestinal lamina propria from CalEnt-infected chickens at 14 dpi via oculonasal route. Scale bar = 20 µm.

**Table 1 viruses-13-01477-t001:** Experimental groups composed of commercial broilers unchallenged or challenged with infectious bronchitis virus strains with enteric (CalEnt) and respiratory (M41) tropisms.

Group	Virus Strain	Infectious Route	No. of Birds at Challenge ^1^
1	CalEnt	Crop gavage	38
2	CalEnt	Oculonasal	39
3	M41	Crop gavage	39
4	M41	Oculonasal	40
5	Uninfected	---	40

^1^ Four birds deceased prior to the virus challenge.

**Table 2 viruses-13-01477-t002:** Percent nucleotide identity between infectious bronchitis virus CalEnt strain and other relevant IBV strains. The highest spike (S) gene homologies are bolded and colored. The genes are sorted from lowest to highest average percent identity.

Gene	Ark DPI	Cal99	Cal 557	Cal 56b	Cal95	Conn46	FL18288	M41	H120	M41vac	GA98	TCoV
**Spike**	**84.6**	**88.4**	83.6	83.2	83.4	80.47	80.2	80.8	80.8	80.8	67.2	47.5
**Envelope**	91.6	90.4	80.2	77.2	79.0	91.89	90.1	91.0	86.5	86.5	88.6	88.0
**3a**	97.1	90.2	93.1	89.7	90.2	90.23	89.7	90.2	85.1	85.1	94.8	89.7
**Matrix**	95.2	95.3	91.6	89.1	89.3	95.15	95.0	95.2	90.5	90.3	89.7	94.3
**3b**	96.9	96.4	91.3	93.3	91.3	97.44	96.4	97.4	83.1	82.8	94.4	95.9
**1a**	94.3	93.8	94.2	95.6	94.7	93.77	93.7	95.6	93.8	93.5	95.6	94.1
**1ab**	95.1	95.2	95.1	96.3	95.3	95.34	95.3	95.8	93.8	93.9	96.0	95.0
**5a**	98.0	98.0	96.7	91.4	89.9	97.98	97.5	98.0	92.4	93.7	97.5	95.0
**Nucleocapsid**	98.7	95.1	94.5	98.7	94.1	99.07	98.7	99.1	93.4	95.2	99.0	95.3
**5b**	98.4	98.0	97.6	98.8	98.8	99.20	99.2	98.4	97.2	97.8	99.2	96.4

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
