# Peer review of "Molecular Biology and Pathological Process of an Infectious Bronchitis Virus with Enteric Tropism in Commercial Broilers"

_viruses, 2021, doi:10.3390/v13081477_

Round 1

Reviewer 1 Report

The authors report the full genome sequence of CalEnt, an Infectious Bronchitis Virus (IBV) strain, and compare the outcomes of experimental infections with the strain to another strain and show that CalEnt has a tropim for the intestinal tract. 
In the methods section, the assembly procedure as to be described in more detail. 
Why was the dosage of CalEnt 10 times higher than the dosage of M41?
Superscripts a and b in Fig. 2 and 3 have to be explained

Reviewer 2 Report

Ana P. da Silva and co-authors investigated the molecular characteristics of an enterotropic IBV previously isolated and partially characterized at their laboratory.  The investigators further studied the tropism of the virus after inoculation through two different routes of commercial chickens and semi-quantified the shedding amount of virus through the cloacal route and in tears.  The studying of unusual variants of IBV is of importance to the field, rendering the manuscript of interest.  However, the presented work has some flaws that need to be addressed prior it being suitable for publication.

One of the main points that need attention is the phylogenetic analysis of the spike protein gene and its hypervariable region. At a minimum, the respective trees need to be redone with a larger dataset of sequences.  This reviewer recommends that the authors download the sequences provided in the supplemental materials of the Valastro et al. IBV classification paper (https://doi.org/10.1016/j.meegid.2016.02.015) and used that dataset to align and build a tree for each - the whole spike protein gene and the hypervariable region.  These will provide the needed resolution for the analysis.

Another major concern is that only reference-based assembly was performed for the complete genome of CalEnt.  Reference-based assemblies are somewhat limited as they force alignment to an existing template and indel may be omitted in the process.  What gap open penalty settings were used in the assembly?  What guarantees that such indels were not missed?  Alternatively, the authors should perform de novo assembly, there are examples of such approaches for RNA viruses in the literature.

No recombination analysis was performed.  The authors should consider using a friendly tool like RDP or similar to perform recombination analysis.  The results that the hypervariable region is very closely related to Cal99 while the whole spike protein gene is not suggest that there may be a recombination event that took place some time in the past.  The authors hold some discussion about the possibility of circulating Cal99 and live vaccines having a role in the evolution of CalEnt but current results do not fully support such statement.  A recombination analysis will provide good evidence if this is the case.  The authors should blast the spike protein gene of CalEnt and download and align all available full gene sequences and run them through the different available recombination tools.

Some minor comments:

L13: there are two 4s

L127: What % NaCl?

L128-130: were there birds sacrificed?  Please provide full details of the design.

L141: reference #20 is in German language.  Would this be easily understandable for all readers?  If not all details are in #21, maybe provide some details of the methods from #20.

L180: What is the cutoff for the assay? It seems that the SD is almost as as high as the average titer.

L184: Within M21-infected groups, at 4 dpi the birds inoculated.....(otherwise it sounds that the second group was inoculated at 4dpi).

L242-273: Too much results repetition

L279-280: ....present respiratory signs.

Table S4: Did the authors build standard curves?  If so, present results as EID50 which is more relevant.  Ct values between viruses are not directly comparable as the efficiency of a set of primers/probe is not the same with all viruses.  For this reason, a standard curve using a titered virus (M41 for M41 groups and CalEnt for CalEnt groups) is more accurate.  Otherwise, please discuss that although not entirely precise, the 44-Cq still gives some general idea of shedding amounts.

Round 2

Reviewer 2 Report

The authors have properly and sufficiently addressed all comments raised by the reviewer.  This reviewer hopes that the authors find their manuscript improved.